# Differentiation of Baboon (*Papio anubis*) Induced-Pluripotent Stem Cells into Enucleated Red Blood Cells

**DOI:** 10.3390/cells8101282

**Published:** 2019-10-19

**Authors:** Emmanuel N. Olivier, Kai Wang, Joshua Grossman, Nadim Mahmud, Eric E. Bouhassira

**Affiliations:** 1Department of Cell Biology, Albert Einstein College of Medicine, Bronx, NY 10461, USA; emmanuel.olivier@einstein.yu.edu (E.N.O.); Kai.wang@einstein.yu.edu (K.W.); josh.a.grossman@gmail.com (J.G.); 2Division of Hematology/Oncology, Department of Medicine, University of Illinois College of Medicine, Chicago, IL 60612, USA; Nadim@UIC.EDU

**Keywords:** red blood cells, non-human primates, engineered red blood cells, induced pluripotent stem cells, enucleation, animal models

## Abstract

As cell culture methods and stem cell biology have progressed, the in vitro production of cultured RBCs (cRBCs) has emerged as a viable option to produce cells for transfusion or to carry therapeutic cargoes. RBCs produced in culture can be quality-tested either by xeno-transfusion of human cells into immuno-deficient animals, or by transfusion of autologous cells in immuno-competent models. Although murine xeno-transfusion methods have improved, they must be complemented by studies in immuno-competent models. Non-human primates (NHPs) are important pre-clinical, large animal models due to their high biological and developmental similarities with humans, including their comparable hematopoietic and immune systems. Among NHPs, baboons are particularly attractive to validate cRBCs because of the wealth of data available on the characteristics of RBCs in this species that have been generated by past blood transfusion studies. We report here that we have developed a method to produce enucleated cRBCs by differentiation of baboon induced pluripotent stem cells (iPSCs). This method will enable the use of baboons to evaluate therapeutic cRBCs and generate essential pre-clinical data in an immuno-competent, large animal model. Production of the enucleated baboon cRBCs was achieved by adapting the PSC-RED protocol that we previously developed for human cells. Baboon-PSC-RED is an efficient chemically-defined method to differentiate iPSCs into cRBCs that are about 40% to 50% enucleated. PSC-RED is relatively low cost because it requires no albumin and only small amounts of recombinant transferrin.

## 1. Introduction

Blood transfusions have been clinically useful for more than 100 years. The idea that RBCs could be engineered to serve as more than just oxygen carriers is almost as old [1]. Drug delivery through therapeutic RBCs compared to direct drug injection in the plasma has generated considerable interest, because this approach shields the drug from the immune system, which decreases the risks of allergy; increases drug half-life, as demonstrated by studies on asparaginase encapsulation inside RBCs [2]; and spatially restricts drugs to the lumen of the cardio-vascular system, which decreases toxicity by limiting diffusion outside of blood vessels [3].

Initial efforts focused on modifying RBCs obtained from volunteers by altering their surface antigens to make them more universal [4], and by decorating them with antibodies or other surface molecules [5,6,7,8,9,10]. These decorated RBCs have been shown to be useful to present antigens, carry therapeutic drugs, or immunize against toxins, to cite just a few applications. RBCs have also been loaded with therapeutically useful content, such as asparaginase or dexamethasone through hypotonic shock and several clinical trials are currently in progress [11,12], attesting to the potential of this technology.

As cell culture methods and stem cell biology have progressed, in vitro production has emerged as an alternative strategy to produce RBCs by differentiation of stem or progenitor cells which can be genetically modified to express therapeutically useful drugs.

One major advantage of in vitro production is that genetically homogeneous cells can be produced from the stem cells of rare donors carrying desirable blood groups that are compatible with a very large fraction of the population. If the source cells are immortal, unlimited numbers of cells can be produced, which eliminates the risk of contamination by unknown or emerging pathogens associated with collection of cells from donors, and decreases production complications associated with the genetic heterogeneity of the donors, which are two drawbacks of the use of multiple donors.

Pluripotent stem cells are a potentially inexhaustible source of cells for industrial production because of their unlimited self-renewal capacity, karyotypic stability, and ease of production. We and others have, therefore, focused on this model to attempt to produce cultured RBCs (cRBCs) for clinical applications [13,14,15,16,17,18,19,20].

We recently described a chemically-defined method to produce cRBCs from iPSCs, termed pluripotent stem cell-robust erythroid differentiation (PSC-RED) [21]. One advantage of PSC-RED compared to other protocols is that the cRBCs produced enucleate at about 40% to 50%, eliminating one of the major road blocks for adoption of this technology for cRBC production. Another advantage of the PSC-RED protocol is that it requires no albumin and only small amounts of recombinant transferrin, which significantly decreases the cost of cRBC production, since these two components represent a large fraction of the cost of previously published protocols.

CRBCs must be quality controlled and clinically validated. Many in vitro methods to characterize the morphological, biochemical, and immunogenic properties of RBCs are available, but testing in animal models remains the gold standard to assess new cell therapeutics prior to human clinical trials.

Cultured RBCs can be tested in animal models either by transfusion of autologous manufactured cells, or by xeno-transfusion of human cells into immuno-deficient animals.

Autologous RBCs transfusions in the murine species are not very informative because cross-strain transfusions can typically be performed without eliciting any adverse transfusion reaction.

Xeno-transfusion in mice is difficult because human RBCs only survive a short time in the mouse circulatory system, even in the absence of all acquired immunity. Although the survival time of human RBCs can be increased by the complete elimination of innate immunity by repeated injections of clodronate liposomes combined with the inhibition of complement activity with cobra venom gactor [22], the toxicity of these compounds, and the complete lack of both immune and innate immunity would greatly limit, in most situations, the power of the model to assess therapeutic efficacy or detect critical manufacturing defects in cRBCs.

Since both xeno-and autologous transfusions in the mouse are associated with major technical difficulties, we have focused on non-human primates (NHPs) (baboon) as potential models to test cRBCs. Due to the high biological and developmental similarity between NHPs and humans, NHPs models offer a great resource to conduct preclinical studies prior to studies in humans.

NHPs embryonic stem cells (ESCs) have been isolated from rhesus monkeys (*Macaca mulatta*), common marmosets (*Callithrix jacchus*) [23], cynomolgus monkeys (*Macaca fascicularis*) [24], African green monkeys (*Cercopithecus aethiops*) [25], and baboons (*Papio Anubis*) [26]. IPSCs have been generated for multiple primate species by the over-expression of pluripotency factors using a variety of reprogramming strategies [27,28,29,30]. All non-human primate ESCs and iPSCs have been shown to share the same pluripotency markers as human ESCs and require bFGF rather than LIF to grow in self-renewal conditions. The availability of these cells has opened the way to use NHP iPSCs to generate pre-clinical data, but protocols to differentiate these iPSCs into mature cell types are generally not available, which limits the uses of these cells.

We report here that we have developed a method to produce enucleated cRBCs from baboon iPSCs. This advance will allow the use of this species to evaluate therapeutic cRBCs and generate essential pre-clinical data in an immuno-competent model.

## 2. Materials and Methods

### 2.1. Reagents

The suppliers for all reagents are provided in Appendix A.

### 2.2. Samples

Mobilized peripheral blood (MBP) CD34+ cells were obtained from healthy baboons under animal protocols approved by the UIC animal institute.

### 2.3. Baboon CD34+ Cell Reprogramming

CytoTune-iPS 2.0 Sendai Reprogramming Kit (Invitrogen, Carlsbad, CA, USA) was used to reprogram baboon peripheral blood CD34+ cells. The day before reprogramming, frozen baboon MPB CD34+ cells were thawed and cultured overnight in StemSpan Serum Free Expansion Medium (StemCell Technologies, Vancouver, BC, Canada) containing SCF + TPO + IGF2 + Flt3-L. 2 × 10^4^ and were transduced in a well of a 96-well flat bottom plate by 1 × 10^5^ CIU of hKOS, 1 × 10^5^ CIU of hc-myc, 6 × 10^4^ CIU of hKlf4, and 4 µg/mL of Polybrene (Millipore, Burlington, MA, USA) in a total of 50 µL of STIF medium. The plate was first centrifuged at 2250 rpm for 90 min at room temperature, and then cultured overnight after an additional 50 µL of STIF medium was added into the well. On the next day of transduction (day 1), the medium with the viruses was replaced by fresh STIF medium.

On day 3 of transduction, the transduced cells were transferred into a 24-well plate coated with an irradiation-inactivated MEF feeder layer with 0.5 mL of StemSpan SFEM medium without cytokines and cultured for 3 days with a medium change on day 5. On day 7 of transduction, the culture medium was changed to half StemSpan SFEM without cytokines and half PluriStem (Millipore) iPSC culture medium, with the transition to complete PluriStem culture medium on day 8. From day 9 to day 21 of transduction, the transduced cells were checked every day for the emergence of iPSC-like colonies. PluriStem iPSC culture medium was changed daily. 

Once the size of the iPSC-like colonies reached 5 mm in diameter, the colonies were mechanically dissected into ~1 mm diameter pieces and plated on either on MEF with PluriStem medium or on vitronectin with baboon E8 medium (bE8). For routine subculturing of baboon iPSCs cultured on MEF, cells were first washed with pre-warmed PBS once and then incubated with 0.5 mM EDTA in PBS at room temperature for 1 min. After removing EDTA, cells were broken into small clusters by gentle pipetting in fresh Pluristem medium and plated on a MEF feeder layer at a 1:4 splitting ratio.

### 2.4. Adaptation and Maintenance of Baboon iPSCs in Feeder-Free Culture Conditions

Six-well plate were coated with 20 ug/mL/well of vitronectin solution in PBS overnight at 4 °C. Baboon iPSC feeder-free culture medium (bE8) was slightly modified from Essential 8 medium [31] by doubling the bFGF concentration (200 ng/mL). Each baboon iPSC colony cultured on MEF at passage 2 was dissected mechanically into ~1 mm diameter pieces and plated into pre-warmed bE8 medium in a well of 6-well plate coated with 20 ug/mL/well of vitronectin. Fresh bE8 medium was changed daily. Only the clones that mostly maintained single-layer undifferentiated iPSC morphology in feeder-free conditions were kept for further expansion and characterization.

The protocol for maintaining baboon iPSCs in vitronectin/E8 conditions was similar to that described by Chen et al. [31] for human cells, except that, as above, the concentration of vitronectin to coat the plate, and the amount of bFGF in the E8 medium, were doubled to, respectively, improve attachment and prevent differentiation. As for human cells, baboon iPSCs were passaged by dissociation in 0.5 mM EDTA but the incubation time had to be reduced to about 1 min, followed by gentle pipetting (instead of about 5 min for human cells). Fragments of iPSC colonies 1 to 2 mm^2^ in size exhibited maximal plating efficiency. Baboon iPSCs were passaged every 3-4 days depending on their confluence stage. Baboon iPSCs at passages 4–8 were used for characterization.

### 2.5. Short version of the differentiation protocol

The differentiation of Baboon iPSCs in erythroid cells was done according to our previously published PSC-RED protocols. Composition of all media and supplements is provided in Appendix A.

On the eve of the differentiation process (day –1), Baboon iPSC colonies (3 days old) were dissociated by the EDTA method, adapted to generate small clumps of cells, and thereafter, plated on vitronectin in E8 at 10 to 20,000 cells/cm^2^.

On the first day of differentiation (day 0), the E8 medium was replaced by IMIT medium with Appendix A.

On the third day of differentiation (day 2), Appendix A dissolved in IMIT medium was added to the culture medium.

On the fourth day of differentiation (day 3), the cells were dissociated with recombinant trypsin and plated at 30,000 cells/cm^2^ in IMIT medium with Appendix A containing the small molecule SB431542.

On the seventh day of differentiation (day 6), the medium was completely replaced by fresh IMIT medium with Appendix A containing the small molecule UM171 instead of SB431542. The cells were re-suspended and plated at 50,000 cells/cm^2^; the cytokines were fully renewed on day 8 and the cell density was maintained at below 150,000 cells/cm^2^.

On the eleventh day of differentiation (day 10), the cells were plated at 0.66 × 10^5^ cells/mL in IMIT medium with the SED supplement. From day 10 to day 17 the cells were maintained under 1.5 × 10^6^ cells/mL and the SED supplement was renewed every two days.

On the seventeenth day of differentiation (day 16), the cells were plated at 2 × 10^5^/mL in IMIT medium with the SER supplement. From day 16 to day 22 the cells were maintained under 1.5 × 10^6^ cells/mL and the SER supplement was renewed every two days.

On the twenty-third day of differentiation (day 22), the cells were plated at 2 × 10^5^/mL in R6 medium with the SER2 supplement. From day 22 to day 28 the cells were maintained under 1.5 × 10^6^ cells/mL and the SER2 supplement was renewed every two days.

On the twenty-ninth day (day 28) after centrifugation the cells were resuspended in R6 medium without any supplement for up to 8 days.

### 2.6. Long version of the differentiation protocol

In the long version of the differentiation protocol, an additional week of expansion was inserted at day 10; the cells were plated at 2 × 10^5^/mL in IMIT medium with Appendix A. During the additional week of culturing, the cells were maintained below 1.5 × 10^6^ cells/mL and the Appendix A was renewed every two days. After this additional step, the differentiation resumes according to the short protocol day 10.

### 2.7. Analysis and Characterization

#### 2.7.1. Teratoma Formation

After baboon iPCS dissociation (80% confluent), the equivalent of one well of a six well plate was suspended in culture medium and mixed with equal volume of matrigel (BD Bioscience, San Jose, CA, USA). The mix was injected intramuscularly into the hind leg of a 6–8 week old NSG (NOD.Cg-Prkdcscid Il2rgtm1Wjl/SzJ) mouse. The tumors visible after 6 to 12 weeks post-injection were sampled, fixed, paraffin embedded, sectioned, and stained with hematoxylin/eosin according standard procedure.

#### 2.7.2. Flow Cytometry

After dissociation and resuspension in staining buffer, baboon iPSCs at passage 4 were stained for 30 min at 4 °C with saturating amounts of PE-conjugated monoclonal antibodies against the pluripotent markers: Stage-specific embryonic antigen 4 (SSEA-4; BD Biosciences, Franklin Lakes, NJ, USA), TRA-1-60, and TRA-1-81 (eBioscience, San Diego, CA, USA). Analysis of the stained cells was performed by flow cytometry on a BD FACSCalibur flow cytometer (BD Biosciences) and the data collected analyzed with the Flowjo software. iPSCs undergoing erythroid differentiation were evaluated by FACS using purified mouse anti-baboon red blood cells from clone E34-73 (BD Biosciences).

#### 2.7.3. Embryoid Body Formation and Immunohistochemistry

For the detection of pluripotent markers by immunohistochemistry, baboon iPSCs were sub-cultured in 2× vitronectin-coated 8-well chamber slides. For the three-germ layer’s differentiation detection, EBs were formed using the hanging drop method and spontaneously differentiated in DMEM supplemented with 20% FBS and L-glutamine in a gelatin-coated chamber slide for 10 days. Cells were fixed with 4% paraformaldehyde for 10 min, permeabilized, and blocked with 0.5% TritonX-100 in PBS with 6% donkey serum for 30 min at room temperature and stained with primary antibodies in 0.1% TritonX-100 in PBS with 6% donkey serum overnight at 4 °C., followed by the incubation of fluorophores’ conjugated secondary antibody for 2 h at room temperature. Labelled slides were counterstained with DAPI and mounted in antifade medium (Prolong, ThermoFisher).

#### 2.7.4. Karyotype

G-banding Karyotyping was performed by the UIC animal facility using standard procedures.

#### 2.7.5. Enucleation

Cells were stained with the DRAQ5 DNA nuclear stain (ThermoFisher, Waltham, MA, USA) and the dead cells were excluded with Propidium Iodide. Analyses of the stained cells were performed by flow cytometry on a BD FACSCalibur flow cytometer (BD Biosciences) and the data collected analyzed with the Flowjo software (Flowjo, Ashland, OR). In some experiments, manual enumeration was used to measure the rate of enucleation (see below).

#### 2.7.6. Cell Enumeration

Cells were counted with a Luna-FL dual channel Automated Cell Counter (Logos Biosystems, Annandale, VA, USA) using acridine orange to visualize the live cells and propidium iodide to exclude the dead cells. Alternatively, the cells were counted manually using a hemocytometer.

#### 2.7.7. Cytological Staining

Erythroid differentiation and enucleation were assessed microscopically by Rapid Romanowsky staining [32] of cytospin preparations using the HEMA-3 kit from Fisher Scientific as recommended by the manufacturer. Cell sizes were estimated on a Zeiss Axiovert 200M microscope using software provided by the manufacturer.

#### 2.7.8. HPLC Analysis

After a double wash with PBS, the cells were lysed in water by 3 quick freeze-thaw cycle. After centrifugation at 16,000 g, supernatants were recovered and stored at −80 °C. The HPLC analysis was performed as previously described [33]. In short, a small volume of lysate diluted in a solution of 40% acetonitrile and 0.18% TFA was briefly filtered and loaded on a VYDAC C4 column (Grace, Columbia, MD, USA). The elution of the globin chains was done with increasing concentration of acetonitrile over 80 min and monitored by measuring O.D. at 220 nm.

#### 2.7.9. Statistical Analysis

GraphPad Prism 7 for Windows was used to perform student t-tests.

## 3. Results

### 3.1. Generation of Baboon iPSCs from Peripheral Blood Cells

CD34+ cells were purified from the peripheral blood of a male adult baboon mobilized with Granulocyte-Colony Stimulating Factor (G-CSF) and frozen. Thawed cells were allowed to recover overnight and transduced with Sendai viruses expressing the four classic reprogramming factors. After 14 days, emerging iPSC colonies were harvested and passaged on plates coated with irradiated mouse embryo fibroblasts (MEFs) in pluriSTEM medium as described by Navara et al. [34].

Reprogramming efficiency was about 0.06% ± 0.0.2% (*n* = 2). Once the lines were established on MEFs, they were highly stable and could be cultured for long periods of time. The morphology of baboon iPSCs is shown in Appendix A.

Efforts to derive baboon iPSCs directly in chemically-defined conditions were not successful, but baboon iPSCs generated on MEFs could be adapted to grow on vitronectin and E8 medium, under the previously-described, chemically-defined culture conditions for human iPSCs [31], by doubling the concentration of vitronectin on the plate and the concentration on FGF2 in the E8 medium. However, even in these optimized conditions, chemically-defined cultures tended to decline after a few passages and could not be expanded for more than a month.

To determine if the iPSC lines that we derived were pluripotent when grown in chemically-defined conditions, we first characterized them by flow cytometry for expression of marker SSEA3, SSEA-4, TRA-1-60, and TRA-1-81. As shown in Figure 1a and Appendix A, SSEA-4, TRA-1-60, and TRA 1-81 were detected on baboon iPSCs, albeit at a lower level than in human cells. SSEA-3, which is difficult to detect in human cells grown in chemically-defined conditions, was undetectable on baboon iPSCs. Levels of expression of SSEA-3, TRA-1-60, and TRA-1-81were similar whether the cells were grown on MEF or in chemically-defined conditions. The expression of SSEA-4 was higher when the baboon iPSCs were grown on MEFs (Appendix A).

To determine if baboon iPSCs maintained in chemically-defined conditions could generate the three germ layers in an in vivo assay, we produced teratomas by intramuscular injections into the hind leg of immuno-deficient mice. Harvesting of the teratomas six to eight weeks post-injection, followed by Hematoxylin & Eosin staining demonstrated that the teratomas contained multiple lineages from all three germ layers (Figure 1b and Appendix A), suggesting that the baboon iPSCs were pluripotent.

To confirm these results, we differentiated the iPSCs into embryoid bodies using the hanging drop method and taking advantage of spontaneous differentiation in the presence of 20% fetal bovine serum. Embryoid bodies thusly generated were stained with antibodies against α-feto-protein, α-smooth muscle, and β III tubulin, which are, respectively, markers for the endodermal, mesodermal, and ectodermal germ layers. As illustrated in Figure 1c and Appendix A, these experiments confirmed that baboon iPSCs could differentiate into lineages representative of the three germ layers. Based on these experiments, we concluded that our baboon iPSCs were pluripotent.

To determine if our baboon iPSCs were karyotypically normal, clones 1 and 3, which had been maintained for about eight passages in chemically-defined conditions, were karyotyped using a standard G-banding procedure. As illustrated in Figure 1d, these experiments demonstrated that the two lines of iPSCs tested were karyotypically normal.

### 3.2. Differentiation into Erythroid Cells

We recently reported that human iPSCs could be differentiated into erythroid cells in albumin-free, chemically-defined conditions using either the short or long PSC-RED protocols [21]. The short-PSC-RED protocol yielded about 300,000 cRBCs per iPSC but the cells produced enucleate poorly. By contrast, the long-PSC-RED protocol, in which iPSC-derived hematopoietic progenitor cells (HPCs) were expanded for an additional week in conditions that were designed to favor expansion without differentiation, cell yields were about two times less, but the cells enucleated at a high rate. To determine if these protocols could be used to produce enucleated baboon cRBCs, we differentiated baboon iPSCs using both the short and the long versions of the protocol (Figure 2a).

Cultures of clones 1 and 3 according to the short protocol yielded, respectively, averages (±S.D.) of 21,756 ± 8187 since it is a relatively trivial matter to start with a higher number of iPSCs. To evaluate the quality of the cells produced, we first analyzed their morphology after rapid Romanowsky staining on various days during differentiation (Figure 3 and Figure 4, Appendix A). As expected, the cells evolved through the different stages of erythroid differentiation in a semi-synchronized manner (Figure 5a).

The cells obtained with the long protocol were delayed compared to the cells from the short protocol, likely because the incubation in the S4 supplement induces the proliferation of the iPSC-derived HPCs with limited differentiation. The large majority of the cells obtained with the short protocol reached the orthochromatic stage after about 28 days of culture but failed to advance to the reticulocyte stage. Instead, they accumulated vesicles of unknown content and eventually died. By contrast, the cells obtained with the long protocol reached the orthochromatic stage at about day 30 to 35 and proceeded to enucleation. To quantify these results, we examined the cells by flow cytometry. Analysis with nuclear stain Draq5 confirmed that the cells obtained with the short protocol did not enucleate and revealed that with the long protocol, the rate of enucleation was high in both clones and averaged 48.8 ± 4.2% (*n* = 4, Figure 5b).

Analysis of the cells produced using the PSC-RED protocol with the E34-731 monoclonal antibody, which specifically reacts with baboon erythroid cells but not with leucocytes or platelets, demonstrated that the percentage of erythroid cells progressively increased, reaching more than 99% at the end of the cultures (Figure 5c).

Globin expression: We have previously shown that human iPSC-derived cRBCs obtained with the short protocol are less developmentally mature than those obtained with the long protocol since they express a higher percentage of embryonic globins and a smaller percentage of fetal and adult globins [21]. Developmental baboon erythropoiesis is very similar to humans [35,36]. The first globin-chains expressed are the ε and ζ-globin. These embryonic chains are replaced by α and γ-globin chains early in gestation and a second globin switch occurs around birth with the γ-globin chains are replaced by β-globin chains.

To evaluate the developmental maturation of iPSC-derived baboon cRBCs, we analyzed globin chain expression by HPLC (Figure 6). This revealed that baboon cRBCs produced by the short PSC-RED protocol expressed a mixture of about 70% ε-globin and 30% γ-globin from the β-like cluster, while cells produced with the long PSC-RED protocol expressed 40% ε-globin, 58% γ-globin, and 2% β-globin demonstrating that the cells produced from the long protocol express a more developmentally mature globin chain mix.

Analysis of the globin chains produced by the α-like globin cluster revealed a similar globin switch (41% α-globin and 59% ζ-globin production in cells obtained with the short protocol, versus 80% α-globin and 20% ζ-globin in cells obtained with the long protocol). Statistical analyses of these results revealed that the differences in globin expression observed between the short and long protocols were significant for all globin genes (*t*-test *p*-values <0.001 in all cases, *n* = 4, Figure 6).

## 4. Discussion

We reprogrammed baboon CD34+ cells from mobilized peripheral blood into iPSCs using the Sendai virus method. The frequency of reprogramming that we observed was lower than what we can achieve with humans CD34+ cells [37], but was adequate to produce many lines of iPSCs because of the large number of available donor cells. Previous attempts with a plasmid transfection method that is effective with human mononuclear cells were not successful (data not shown). These results are similar to reports by Navara et al. who have also observed low reprogramming efficiency of baboon fibroblasts and blood cells using either a retroviral or a Sendai virus methods [27]. Multiple types of baboon cells are, therefore, difficult to reprogram with multiple methods of over-expression of the reprogramming factors. This suggests that the limiting factor might be suboptimal cell culture conditions during the reprogramming process rather than the reprogramming factor’s delivery method. Optimizing culture conditions might improve reprogramming efficiency in the future.

We have shown that baboon iPSCs can be maintained for about a month in chemically-defined conditions using minor modifications of the protocol described by Chen et al. [31] for human iPSCs. However, these chemically-defined conditions were not appropriate for the generation of iPSCs.

We have demonstrated that the PSC-RED protocols allow the production of enucleated baboon cRBCs. As described for human iPSCs, the type of cRBCs produced can be modulated by altering the length of the protocol. The short protocol yields cells that are similar to primitive erythroid cells. The long PSC-RED protocol yields cells that are more developmentally mature and that resemble definitive fetal erythroid cells. As with human cells, even with the long protocol, expression of the β-globin gene is low compared to adult cells which express close to 100% β-globin.

The yield of cRBCs produced was lower than for human cells and likely could be improved through further optimization of the concentration of cytokines used during the differentiation or by using baboon cytokines if available. Nevertheless, the yield obtained was high enough to allow the production of enucleated cRBCs in sufficient amounts to perform pre-clinical trials in this species.

Among the NHP species, baboons and Rhesus monkeys are the most frequently used for medical research purposes. Both species are about equally distant to humans, but baboons have specific advantages for some applications, including a size and an anatomy that is more similar to that of humans [38]. Of particular interest for cRBCs research, baboons have been used extensively to study blood transfusion and there is a wealth of data available on the characteristics of RBCs in this species [39].

One of the major advantages of using iPSCs is that they can be easily genetically engineered using precise, site-specific insertion methods. Because of the phylogenetic proximity between baboons and humans, expression constructs designed for the latter species can be used in the baboon with minor modifications and are expected to be expressed in a very similar manner in both species. We successfully transfused ABO-matched blood as supportive care after partial or complete myelo-ablation in a baboon [40]. ABO matched or autologous transfusion of baboon cRBCs genetically engineered to carry therapeutics might, therefore, provide a platform to rapidly test therapeutic cRBCs in an immuno-competent setting.

## Figures and Tables

**Figure 1 cells-08-01282-f001:**
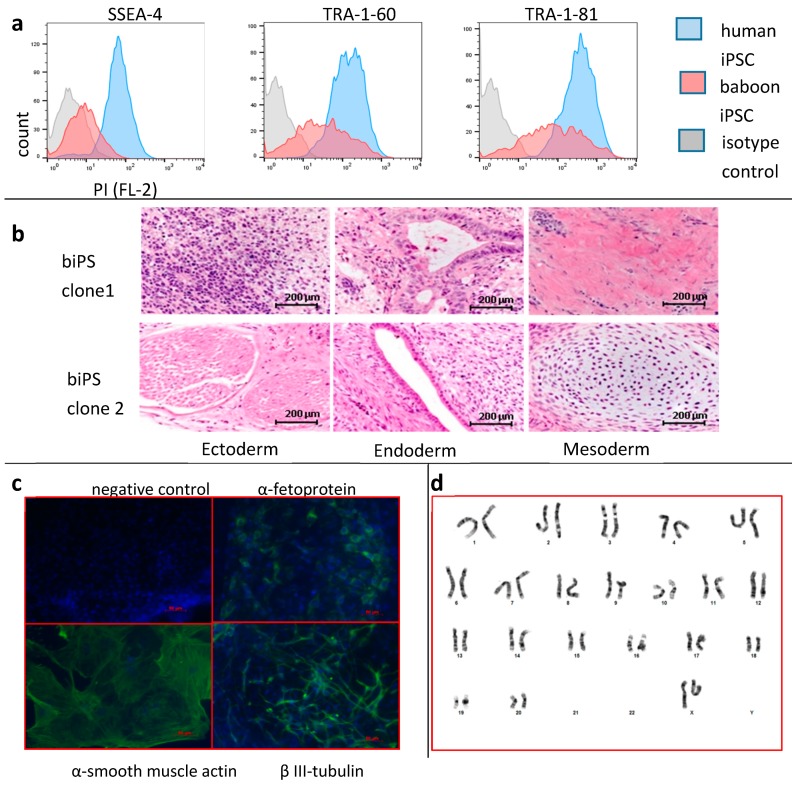
Production of baboon iPSCs: (**a**) FACS analysis of human and baboon iPSCs grown in chemically-defined-conditions. Blue histogram: human iPSCs; red histograms: baboon iPSCs; grey histograms: isotype controls. Baboon iPSCs express pluripotency markers albeit at lower levels than human iPSCs. (**b**) Teratoma analysis. 1 × 10^6^ baboon iPSCs were injected intramuscularly into the hind leg of a 6–8 week old NSG mouse. Six weeks later, tumors were fixed in 10% formalin, paraffin embedded, sectioned, and stained with hematoxylin/eosin. Tumors from two different iPSC clones are shown. Structures originating from all three germ layers were found in most tumors analyzed; (**c**) embryoid bodies were formed using the hanging drop method in 20% FBS for 10 days. Cells were fixed with paraformaldehyde, stained with indicated antibodies, and counterstained with DAPI. Cells expressing α-feto-protein (endoderm), α-smooth muscle actin (mesoderm) and β-III-tubulin were detectable in 10-day EBs. Baboon iPSCs maintained in chemically-defined conditions are pluripotent; (**d**) karyotyping: two clones of iPSCs were analyzed using standard karyotyping methods.

**Figure 2 cells-08-01282-f002:**
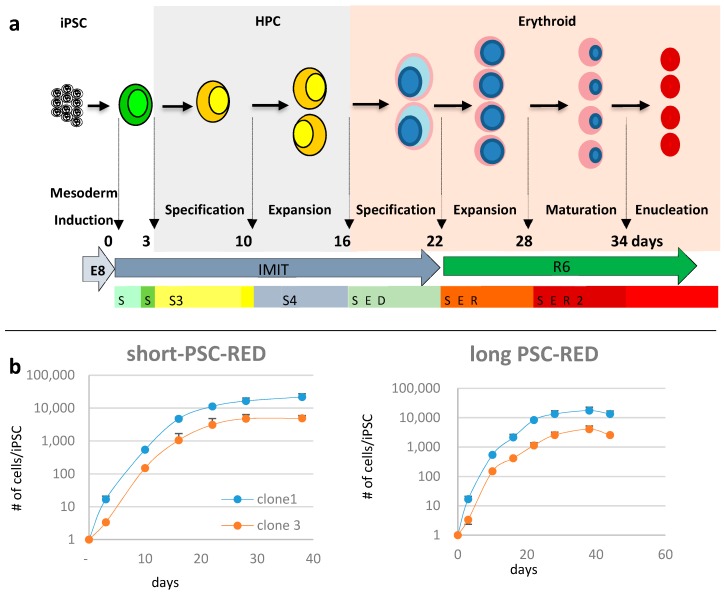
PSC-RED protocols: (**a**) Scheme illustrating the long PSC-RED protocol to produce enucleated cultured RBCs (cRBCs) from iPSC. In the short version of the protocol, the expansion step in S4 from day 10 to day 16 is omitted. Media and cytokine supplements are described in the Methods section; (**b**) graphs illustrating the number of cRBCs/iPSCs observed during the erythroid differentiation of two different baboon iPSC clones (clone 1 and clone 3) using the short or long PSC-RED protocols. Data are expressed as average ± SEM of 3 independent experiments.

**Figure 3 cells-08-01282-f003:**
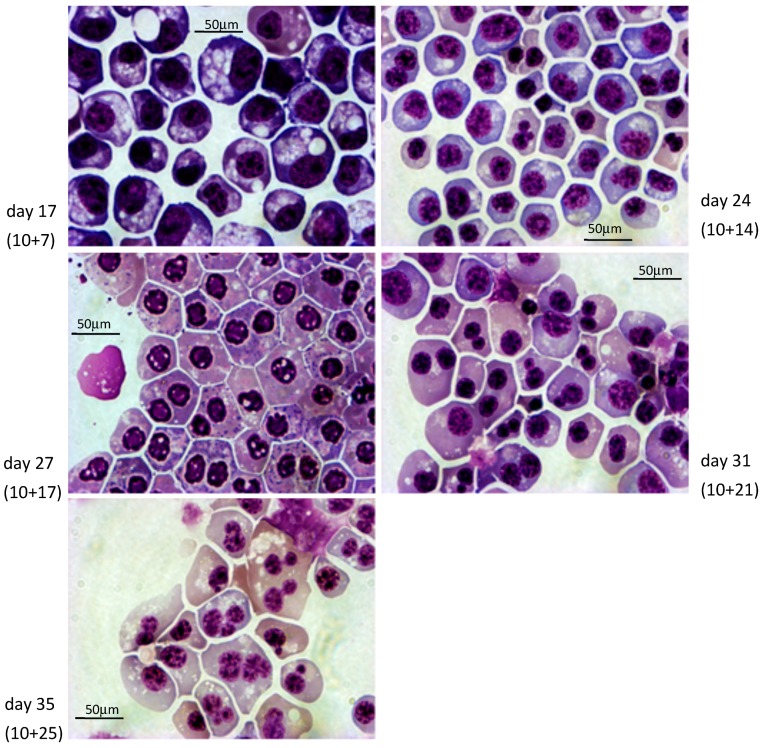
Morphological analysis of baboon iPSCs differentiating into cRBCs using the short PSC-RED protocol. 630x magnification micrograph illustrating the typical morphology of the erythroid cells observed at days 17, 24, 27, 31, and 35 of culture after rapid Romanowsky staining. The majority of cells were pro or basophilic erythroblasts at day 17, and orthochromatic erythroblasts at day 35.

**Figure 4 cells-08-01282-f004:**
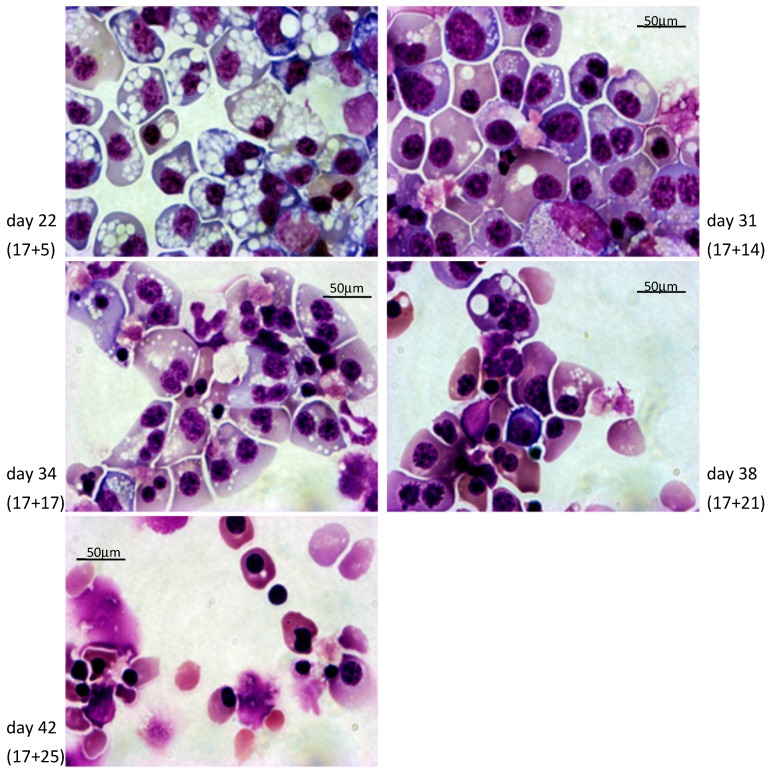
Morphological analysis of baboon iPSCs differentiating into cRBCs using the long PSC-RED protocol: 630x magnification micrograph illustrating the typical morphology of the erythroid cells observed at days 22, 31, 34, 38, and 42 of culture after rapid Romanowsky staining. The majority of cells were pro-erythroblasts at day 22, and orthochromatic erythroblasts and reticulocytes at day 42.

**Figure 5 cells-08-01282-f005:**
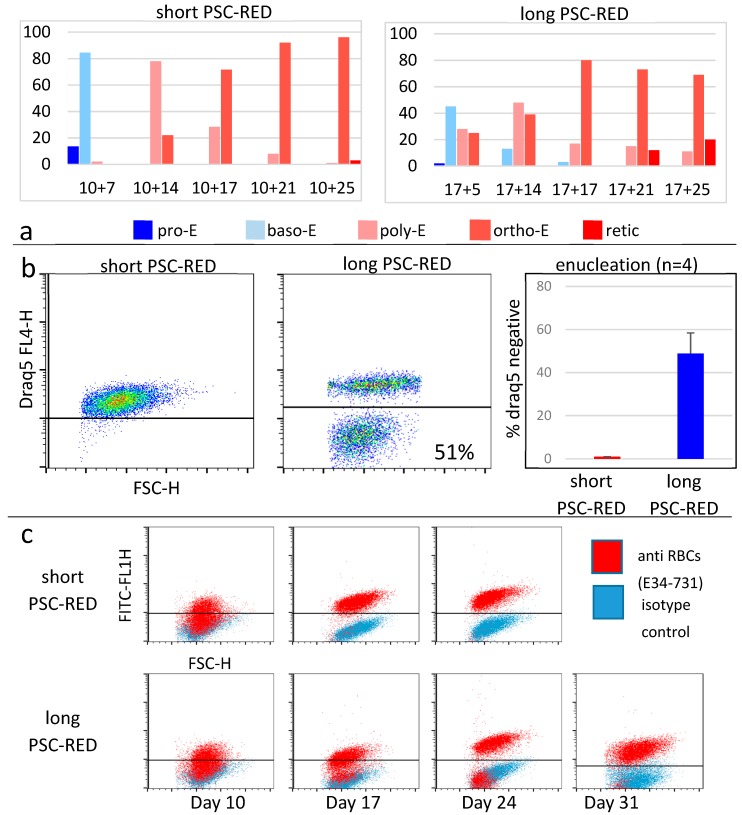
Differentiation analysis: (**a**) Cells were spun on microscope slides, stained using the rapid Romanowsky methods and classified as proE (pro-erythroblasts); basoE (basophilic erythroblasts); polyE (polychromatophilic erythroblasts); orthoE (orthochromatophilic erythroblasts) or retic. (reticulocytes (enucleated cRBCs)). Graphs illustrate the average (±SD) number of erythroid precursors during differentiation using either the short (left) or long (right) PSC-RED protocols (*n* = 2); (**b**) FACS plots representative of the enucleation rates as determined by DNA content measurement using Draq5. The rate of enucleation is much higher using the long protocol (Student’s *t*-test *p*-value between the rate of enucleation short and long protocol in s: <0.00001) (**c**) FACS plots illustrating the proportion of erythroid cells during the culture using anti-baboon RBCs antibody E34-731.

**Figure 6 cells-08-01282-f006:**
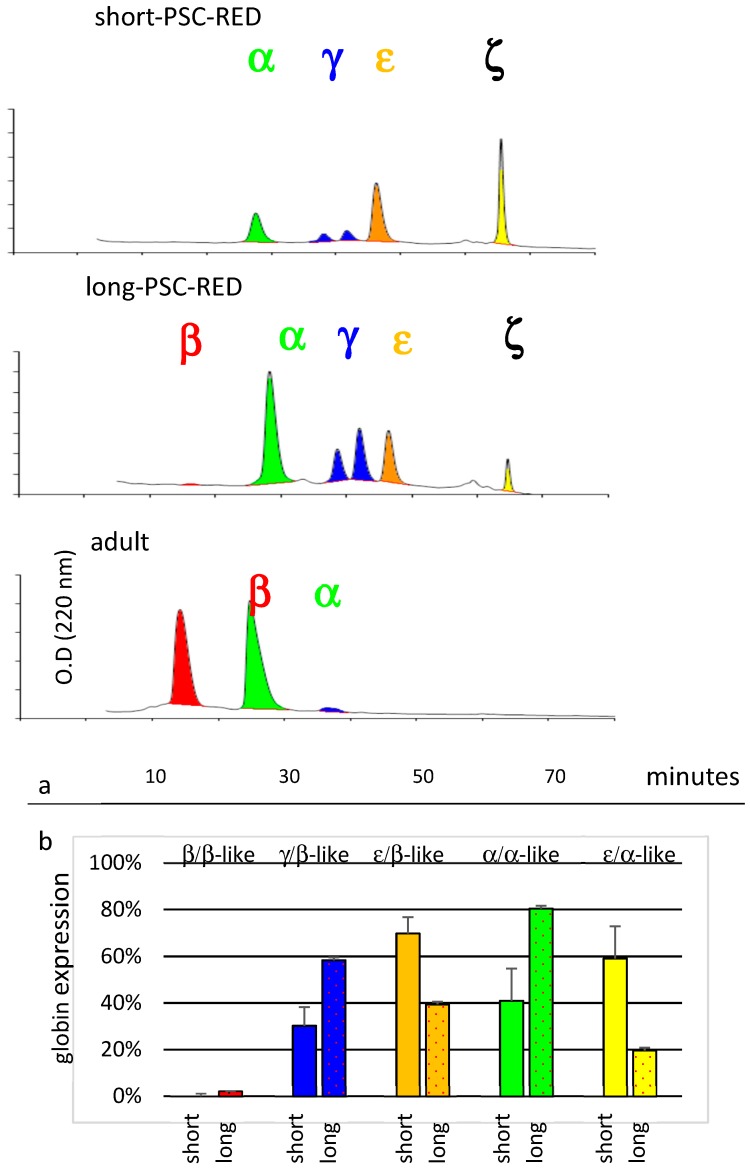
Globin expression. (**a**) HPLC chromatograms illustrating globin expression. Cultured RBCs obtained at day 32 (short-protocol) or 36 (long-protocol) were lysed and analyzed by HPLCs. RBCs from an adult baboon were used as controls (lower chromatogram). ζ and ε-globins are expressed at lower levels in cells obtained with the short protocol; (**b**) histograms summarizing the results of the HPLC analysis. Percentage of a-like = 100 × (α or ζ)/(α + ζ). Percentage of β-like = 100 × (ε( Gα + Aγ) or β)/(ε + Gγ + Aγ + β). Data are the average ± SEM of four independent experiments. The long PSC-RED protocol yields more developmentally mature cells.

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
