# Peer review of "Differentiation of Baboon (Papio anubis) Induced-Pluripotent Stem Cells into Enucleated Red Blood Cells"

_cells, 2019, doi:10.3390/cells8101282_

Round 1

Reviewer 1 Report

Line 12 and 61 : „cells are immortal”. I suggest more secure expression, such as: “unlimited self-renewal capacity”,  “infinite proliferative capacity”.

Abstract and/or Introduction: I also suggest underline the main aim/objective of this study

Methods: Authors used the integration-free method based on Sendai virus. Perhaps, this is the reason of relatively low efficiency of reprogramming process (low expression of pluripotency markers- Fig. 1) ? Could you comment on that?

Fig. 1A: If I correctly understood the presented results are generated with the participation of iPSCs cultured in chemically-defined condition. Maybe there is another reason explaining the high heterogeneity and relatively low efficiency of reprogramming process (low expression of pluripotency markers)? Did the authors compare iPSCs cultured in both different conditions (e.g. the capacity to form 3 germ layers) not only the expression of some markers (lines: 296-297)?

Could you add picture of iPSCs colonies to evaluate their morphology?

Did the authors consider the fluorescence-activated cell sorting of iPSCs to obtain more homogenous population with higher expression of pluripotency markers for further differentiation?

Fig. 1C The quality of photos should be higher.

What is the process of “spontaneous” enucleation acquiring during iPSC differentiation?

The authors used the albumin-free, low-transferrin media to reduce costs. However, it important to point out that the use of all described reagents and growth factors does not ensure the cost-effective approach. Did the authors try to limit the amount (or concentration) of used growth factors?

The authors previously published differentiation protocol with the use of hiPSCs elsewhere. I have the impression that the repetition of this protocol on animal is a small step backwards. Also the used methods and obtained results are notable cut in comparison with author’s previous study. Could you comment on that?

Line 293: TRA-1-81 should be instead of “TRA-1-82”.

Figure 3/Fig 4b (and methods): What statistical methods were used (in according to lines: 260-261 it was student t-test)?  Where is presented the statistically significance?

Did the authors use any positive/negative controls during their staining procedures?

The authors presented results that involves the cytological staining and HPLC analysis. Where are analyses involving characteristic markers for RBCs (e.g. http://www.antibodybeyond.com/reviews/cell-markers/erythrocyte-marker.htm and Fig. E4 B- you previous work). Did the authors perform some functional analyses it they are routinely used?

Discussion section must be improved! It should include the debate involving author’s research as well as other available literature data.

The Conclusion section should be added.

Reviewer 2 Report

In their manuscript, Olivier et al. demonstrate the ability to generate cultured red cells from induced pluripotent stem cells in a non-human primate (Baboon) model.

As with their previous publication using the same method in human IPSC to red cell cultures (Olivier et al. Experimental Hematology, 2019), they demonstrate significantly higher than previously reported enucleation rates using their method. In the Baboon model, up to a 50% enucleation rate was observed. As with the human model, no albumin and only small amounts of recombinant transferrin were used in cultures, bringing down production costs and moving one step closer towards clinical translatability.

While several groups have successfully generated IPSC from non-human primate models, this is the first to the reviewer’s knowledge that red cells have been successfully cultured from non-human primate derived IPSC, making this an important pre-clinical step towards testing survival and other functional aspects of these cells prior to clinical translation.

One caveat of the method is the significantly lower red cell expansion efficiency in the baboon model compared to human IPSC derived or MNC derived cultured red cells. However, for proof-of-concept studies in large animals, the expansion efficiency and total numbers of RBC generated are less critical than generating enough cells for phenotypic and functional testing.

As a concept presently, while the overall expense of generating these cells is significantly high for day to day clinical practice, it should be a considered a “cellular therapy product for special applications”- including for transitory use in highly alloimmunized sickle cell disease patients, for instance, during their bridge period prior to transplantation or gene therapy. Viewed from this perspective, one may be able to justify costs of cultured red cell production at this time for early phase studies as adjuncts to other curative cell and gene therapies.

Reviewer 3 Report

This manuscript describes that iPS cells were established from baboons and induced to differentiate into red blood cells. This research is important in preclinical research using large animals in regenerative medicine. However, there are insufficient points regarding the following points, in particular, data relating to characterization of iPS cells.

1.Fig1b: The quality of the image is low, and it is difficult to judge whether it is differentiated into each germ layer. I would like to know the details of which structures are considered to be germ layers. As an indicator of pluripotency, gene expression data by PCR of embryoid bodies seems to be sufficient.

2.Fig1c: Image quality is low and no scale bar is found. It is not clear whether it is a section.

3. The morphology of the established iPS cells is not shown in the photograph. This is important information and should include a picture of the original cell.

4. Since these iPS cells are negative for SSEA4, is it possible to say that these cells are in the Primed state?

Round 2

Reviewer 1 Report

This manuscript is suitable for publication.